# Effect of Lactic Acid Bacteria on the Lipid Profile of Bean-Based Plant Substitute of Fermented Milk

**DOI:** 10.3390/microorganisms8091348

**Published:** 2020-09-03

**Authors:** Małgorzata Ziarno, Joanna Bryś, Mateusz Parzyszek, Anna Veber

**Affiliations:** 1Division of Milk Technology, Department of Food Technology and Assessment, Institute of Food Science, Warsaw University of Life Sciences—SGGW (WULS–SGGW), 02-787 Warsaw, Poland; 2Department of Chemistry, Institute of Food Science, Warsaw University of Life Sciences—SGGW (WULS–SGGW), 02-787 Warsaw, Poland; joanna_brys@sggw.edu.pl; 3Institute of Horticultural Sciences, Warsaw University of Life Sciences—SGGW (WULS–SGGW), 02-787 Warsaw, Poland; mateuszparzyszek95@gmail.com; 4Department of Food and Biotechnology, Faculty of Agrotechnological, Federal State Budgetary Educational Institution of Higher Education, Omsk State Agrarian University Named after P. A. Stolypin, Instituskaya Area 2, 644008 Omsk, Russia; anna.web@mail.ru

**Keywords:** *Lactobacillus*, lactic acid fermentation, bean–based beverages, fatty acids profile, positional distribution of fatty acids in triacylglycerols

## Abstract

Biological processes of legumes may change their nutritional value of lipids, but there is no research on the fatty acid profile and their position distribution in fermented beverages obtained from germinated bean seeds. The present study aimed to determine the effect of fermentation by *Lactobacillus* strains on the fatty acid profile and their positional distribution in triacylglycerols of beverage obtained from germinated bean “Piękny Jaś Karłowy” (*Phaseolus vulgaris*) fermented by *Lactobacillus* strains. The population of lactobacilli (the pour plate method), pH, the fatty acid profile (gas chromatograph with a flame ionization detector), and the positional distribution of fatty acids in triacylglycerols (GC-FID) were determined before and after the fermentation of received beverages. The fermentation of beverages did not change the lactobacilli population (over 7 log_10_ CFU/g), but changed pH (to 4.7–3.7 or 5.8–5.6), fatty acid profile, and the positional distribution of fatty acids were observed. The fermentation process contributed to an increase in the share of palmitic, stearic, and oleic acids in the fatty acid profile compared to that in raw bean seeds. The fermentation processes changed the share of individual acids in positions sn–1 and sn–3 depending on *Lactobacillus* strain used. Compared to non-fermented beverages, in most fermented beverages, a lower share of palmitic and stearic acids, as well as a higher share of oleic acid in the sn–2 were observed.

## 1. Introduction

There are many fermented foods easily digestible and rich in nutrients. Several of them are suitable vegan and vegetarian diets. Modern technologies offer vegan or vegetarian products obtained from various plant raw materials. One such raw material is beans, a legume plant belonging to the family *Fabaceae*. Bean seeds are a valuable source of easily digestible proteins, carbohydrates, minerals, and various vitamins (e.g., B vitamins). Subjecting bean seeds to biological processes such as germination and fermentation may improve their nutritional value by changing the content or digestibility of certain nutrients (e.g., proteins or polyphenols) or by eliminating undesirable components (e.g., trypsin inhibitors, stachyose, and raffinose). There are few studies on the influence of germination and fermentation processes on the profile of fatty acids and their positional distribution in lipids of bean seeds. Although the content of lipids in the bean seeds is low, i.e., approximately 1–2%, lipids contained in legume seeds are recommended in daily diet because of their high content of polyunsaturated fatty acids (PUFA), which contribute to lowering of blood cholesterol levels [1]. Legumes contain a favorable composition of unsaturated fatty acids such as linoleic acid (C18:2), the share of which ranges from 21 to 53% of the total fatty acid pool. There is also a significant amount of α-linolenic acid, which is necessary for the synthesis of the remaining PUFA from the omega-3 family. The share of α-linolenic acid in the seeds of legumes ranges from 4 to 22% of the total fatty acid pool. The seeds of all types of legumes have a very low share of trans fatty acids, which is less than 1% of the total fatty acid pool [2].

An increasing number of studies is being conducted on the profile of fatty acids in seeds of various legumes [1,3,4,5,6,7,8]. Abdel-Rahman et al. [3] investigated the effect of the use of various technological procedures (such as soaking, germination, cooking, soaking-cooking, and germination-cooking) on the fatty acid profile of mung bean seeds. The obtained results show that all these processes contribute to increase in the share of saturated fatty acids and reduction in the share of unsaturated fatty acids. Changes in the fatty acid profile of mung beans were also investigated by Megat-Rusydi et al. [8], who studied other legumes, namely kidney beans, soybeans, and peanuts. In this study, after 48 h of germination, an increase in the percentage of saturated fatty acids was found in all the tested legumes (kidney beans: from 47.96% of the total fatty acid pool in the control to 51.8% of the total fatty acid pool after germination; mung beans: from 44.78% of the total fatty acid pool in the control to 46.26% of the total fatty acid pool after germination; soybeans: from 16.65% of the total fatty acid pool in the control to 18.62% after germination; peanut: from 20.36% of the total fatty acid pool in the control to 23.43% of the total fatty acid pool after germination). The share of polyunsaturated fatty acids increased only in mung beans (from 43.86 to 50.10% of the total fatty acid pool), while, in other legumes, there were smaller or greater reductions in the percentage of PUFA pool (kidney beans: from 42.31% of the total fatty acid pool in controls to 27.81% of the total fatty acid pool after germination; soybeans: from 62.95% of the total fatty acid pool in controls to 53.7% after germination; peanut: from 37.41% of the total fatty acid pool in controls to 36.35% after germination) [8].

Currently, there is less research work on the subject of research on the influence of various biological and technological processes on the positional distribution of fatty acids in triacylglycerols (TAG) of bean seed lipids. Regarding the positional distribution of TAG fatty acids in Adzuki bean seeds, it was observed that most (>96%) unsaturated fatty acids were accumulated in the sn–2 position of TAGs. Saturated acids tended to accumulate in the sn–1 and sn–3 positions. The exception was oleic acid, which accumulated evenly in all three positions [1]. In broad beans, a significant amount of unsaturated fatty acids (>95%) accumulated in the sn–2 position. Oleic acid alone almost uniformly occupied the sn–1, sn–2, and sn–3 positions. Saturated acids such as palmitic and stearic acids accumulated in the sn–1 and sn–3 positions. An almost identical distribution occurs in peas, in which more than 90% of unsaturated fatty acids are found in the sn–2 position and saturated acids are mainly found in the sn–1 and sn–3 positions [6,7]. There is no research at all on the fatty acid profile and their position distribution in lipids extracted from bean-based fermented beverages obtained from germinated seeds.

The present study aimed to determine the effect of lactic acid fermentation by pure monocultures from the genus *Lactobacillus* on the fatty acid profile and their positional distribution in TAGs of a fermented bean-based beverage obtained from germinated white bean “Piękny Jaś Karłowy” (*Phaseolus vulgaris*).

## 2. Material and Methods

### 2.1. Bacterial Strains and Culture Conditions

The following strains of lactobacilli were used to obtain the bean-based plant substitute of fermented milk: *Lactobacillus delbrueckii* subsp. *bulgaricus* ATCC 11842, *Lactobacillus delbrueckii* subsp. *lactis* ATCC 4797, *Lactobacillus fermentum* ATCC 9338, *Lactobacillus plantarum* DSM 9843, *Lactobacillus brevis* L342 (from the WULS-SGGW pure culture collection), *Lactobacillus acidophilus* La3 (pure cultures from Sacco srl., Codarago, Italy), *Lactobacillus paracasei* BGP1 (pure cultures from Sacco srl.), *Lactobacillus casei* 01 (pure cultures from Chr. Hansen Poland, Czosnów, Poland), *Lactobacillus rhamnosus* LH32 (pure cultures from Chr. Hansen Poland), and *Lactobacillus helveticus* LH-B01 (pure cultures from Chr. Hansen Poland). 

The strains were stored at −80 °C in a 15% solution of glycerol stock. Before application, the strains were systematically transferred (2% *v*/*v*) into sterile De Man-Rogosa-Sharpe broth (Merck KGaA, Darmstadt, Germany). The strains were cultured overnight (anaerobically) at 37 °C.

### 2.2. Preparation of Bean-Based Plant Substitute of Fermented Milks

Bean-based beverages were obtained from the germinated bean seeds of white bean “Piękny Jaś Karłowy” (*P. vulgaris*) purchased from the domestic market in 2018–2019. The process of germination of bean seeds and the preparation of bean-based beverages were performed according to Ziarno et al. [9]. Briefly, the germination process was performed at 25 °C for 3 days. The germinated bean seeds were then mixed with drinking water in a ratio of 1:9 and homogenized until a homogeneous mass was obtained. The bean mass thus obtained was boiled for 60 min to perform the starch gelatinization process and then subjected to the sterilization process at 121 °C for 15 min to inactivate microorganisms and native bean enzymes. Samples of 100 mL of bean-based beverages were inoculated by overnight cultures of individual *Lactobacillus* strains (previously cultured in De Man–Rogosa–Sharpe broth and washed three times using a sterile saline solution) to obtain a cell population of approximately 1 × 10^8^ CFU/g. The inoculated samples were then incubated at 37 °C for 18 h, and after completion of the fermentation process, the samples were cooled to 5 °C by being kept in cold ice water for 30 min. All of these procedures were performed under aseptic conditions.

### 2.3. Microbiological Analysis

Microbiological analysis included the determination of the cell number of *Lactobacillus* strains in the bean-based plant substitute of fermented milk by the pour plate method with two parallel independent replicates of each of the analyzed beverage. The De Man–Rogosa–Sharpe agar (Merck KGaA) was used for the analysis. The Petri dishes with inoculations were incubated at 37 °C for 72 h under anaerobic conditions using an anaerobic jar equipped with Anaerocult A (Merck KGaA). After the completion of incubation, the grown colonies were counted. For colony counting, the Petri dishes in which the number of grown colonies was in the range of 30–300 colonies were chosen. The final results were converted to the number of colony-forming units per gram of the product (CFU/g), and then converted to logarithm (log_10_ CFU/g).

### 2.4. pH Determination

The pH was measured during the fermentation process of bean-based beverages. The pH values of samples were measured directly using the pH meter CP–505 (Elmetron, Zabrze, Poland). The analysis was performed in four replications. The result was read with an accuracy of 0.01 pH unit. On the basis of measurements of pH acidity, curves of acidification of bean-based beverages by individual strains of *Lactobacillus* were constructed.

### 2.5. Lipid Extraction

The samples of raw bean seeds of white bean “Piękny Jaś Karłowy” (*P. vulgaris*) (ground previously) and the samples of bean-based beverages (before and after fermentation) were analyzed for the profile of fatty acids and their positional distribution in TAGs. For this purpose, the samples were extracted using the Folch method [10]. Briefly, 100 mL of chloroform/methanol mixture (in a 1:1 ratio) was added to an aliquot of 75 g of ground raw bean seeds of white bean “Piękny Jaś Karłowy” or 120 g of a bean-based beverage and mixed for 3 min. The mixture thus prepared was placed at 40 °C for 20 min and then cooled, and 100 mL of chloroform was added and vigorously stirred for the next 3 min. The resulting mixture was filtered through Whatman filter 56 paper No. 4 (Whatman plc, Maidstone, UK). Next, 70 mL of 1 M KCl was added to the obtained filtrate and left at 4 °C for overnight. Subsequently, the filtrate was warmed to room temperature and transferred to a separating funnel to separate fractions. The lower layer (chloroform fraction) was collected in a separate vessel for further analysis. A small amount of anhydrous sodium sulfate was added to the obtained chloroform layer and left for 40 min in the dark, with occasional stirring. Subsequently, the sample was again filtered through a filter paper into a previously weighed round-bottom flask. The filtrate was subjected to an evaporation process at 40 °C and then dried under nitrogen atmosphere. After this process, the flask was weighed to determine the content of the lipid fraction. The obtained lipid fraction was dissolved in 5 mL of the mixture (containing 2 mL of hexane, 2 mL of KOH, and 1 mL of methanol) and collected in separate tubes. Next, 2 mL of hexane and 2 mL of KOH methanol mixture (1 mol/1000 mL) were added to the lipid fraction of previously prepared samples in tubes, mixed, and then incubated at 40 °C for 20 min. The samples were then allowed to cool down for 5 min. Approximately 1 mL of the resulting upper phase was transferred to a new tube, and a pinch of anhydrous magnesium sulfate was added. The prepared samples were analyzed by gas chromatography.

### 2.6. Analysis of the Fatty Acid Profile

The samples were analyzed using a 6000 Series gas chromatograph system YL6100 GC with an installed flame ionization detector (Young Lin Instrument Co., LTD, Anyang, Korea) and a MEGA-10 capillary column (MEGA S.r.l., Legnano, Italy) with the following parameters: internal diameter 0.25 mm, film thickness 0.25 μm, length 60 m. The carrier gas used for the samples was nitrogen. The initial oven temperature was set to 70 °C and held for 30 s. The temperature was then increased from 70 to 160 °C at the rate of 15 °C/min. A subsequent temperature increase from 160 to 200 °C occurred at the rate of 1.1 °C/min; the temperature of 200 °C was then maintained for 12 min. The temperature was then further increased from 200 to 225 °C at the rate of 30 °C/min. The end temperature of 225 °C was held for 1 min. The detector temperature was programmed at 250 °C and the injector temperature at 225 °C. The total analysis time for a single sample was 50 min.

Individual fatty acids were detected on the basis of the analysis of retention times, which were compared with a suitably selected external standard (Fatty Acid Methyl Ester Standards, Merck KGaA). The share of individual fatty acids in the fatty acid profile was estimated by determining the area of the peaks in the chromatogram.

### 2.7. Analysis of the Positional Distribution of Fatty Acids in Triacylglycerols

The hydrolysis of TAGs was carried out according to the method described by [1,11] with some modification. Briefly, to perform the hydrolysis of TAGs contained in the lipid fraction of the tested samples, lipase from porcine pancreas (Merck KGaA) was used, which shows activity against ester bonds in the sn–1 and sn–3 positions. For this purpose, 1 mL TRIS–HCl, 0.1 mL CaCl_2_ (2.2%), and 0.25 mL bile salts (0.05%, Merck KGaA) were added to 0.1 g of the sample of the analyzed lipid fraction. The resulting mixture was stirred for 30 s, and 20 mg of lipase from porcine pancreas (Merck KGaA) was then added and mixed again for 30 s. The samples were then incubated in a 40 °C water bath for 3 min, and 1 mL of 6N HCl and 4 mL of diethyl ether were then added and stirred for 1 min. The samples were then centrifuged for 5 min at 4000 rpm. The resulting upper ether layer was transferred to a new tube and dried under nitrogen atmosphere to a volume of approximately 0.5 mL. The samples were then applied to previously prepared chromatography plates (Silica on TLC Alu foils, Fluka Analytical Sigma-Aldrich, St. Louis, MO, USA) and placed in a chromatography chamber with a developing solution (hexane:ether:acetic acid).

After developing the chromatogram and verifying the hydrolysis, the gel was scraped at the level of the monoacylglycerol (MAG) fraction and placed in new test tubes into which 2 mL of diethyl ether was added and stirred for 1 min. The tubes were then centrifuged for 5 min at 4000 rpm. The separated upper ether layer was transferred to new tubes and dried under nitrogen atmosphere; it was then dissolved in 5 mL of the mixture (containing 2 mL of hexane, 2 mL of KOH, and 1 mL of methanol) and collected in separate tubes. Next, 2 mL of hexane and 2 mL of KOH/methanol mixture (1 mol/1000 mL) were added to the lipid fraction samples previously prepared in tubes, mixed, and then incubated at 40 °C for 20 min. Subsequently, the samples were allowed to cool down for 5 min. Approximately 1 mL of the resulting upper phase was transferred to a new tube, and a pinch of anhydrous magnesium sulfate was added. The samples prepared were analyzed by gas chromatography as described above.

The share of fatty acids in the sn–1 and sn–3 positions was calculated using the following Formula (1):(1)sn–1,3 =[3 ×(FA in TAGs)−(FA in sn −2 MAG)]/2
where:

sn–1,3—the share of a fatty acid in the sn–1 and sn–3 positions [%],

(FA in TAGs)—the share of a fatty acid in the starting triacylglycerols [%],

(FA in sn–2 MAG)—the share of a fatty acid in the sn–2 position in the final monoacylglycerols [%].

The composition of the sn–2 fatty acids compared to the total specific fatty acid share in all positions was determined using the following Formula (2):(2)sn–2 =(FA in sn −2 MAG) ×100%/[3 ×(FA in TAGs)]
where:

sn–2—the share of a fatty acid in the sn–2 position [%],

(FA in sn–2 MAG)—the share of a fatty acid in the sn–2 position in the final monoacylglycerols [%],

(FA in TAGs)—the share of a fatty acid in the starting triacylglycerols [%].

### 2.8. Statistical Analysis

All results were reported as arithmetic average and standard deviation calculated from three replicates. Statistical analyses were performed using Statistica 13.1 software (StatSoft, Krakow, Poland). Analysis of variance (ANOVA) was used to determine differences between the mean scores. Differences between the mean values obtained by ANOVA were made using Tukey’s comparison test (at α = 0.05).

## 3. Results and Discussion

### 3.1. Lactic Acid Bacteria Population

All the tested strains were inoculated into the germinated bean beverages with a cell count of 7.5–8.7 log_10_ CFU/g. The statistical analysis showed that the level of the initial lactobacilli population did not depend on the strain (Table 1). Because of fermentation of bean-based beverages obtained from germinated bean seeds, the population of the studied lactobacilli did not change significantly (*p* > 0.05); however, changes in pH and changes in the lipid fraction (discussed below) indicates that bacterial cells showed significant enzymatic activity at that time.

Our previous experiments [12] proved that nonfermented or fermented plant beverages (soy, rice, and coconut-based) can be carriers of live lactic acid bacteria. However, the chemical composition and acidity of beverages can significantly affect the viability of bacterial cells and maintenance of the required minimum population of bacteria [13,14]. Adequate viability of bacterial cells is the first and most important criterion to ensure high quality of plant-based beverages. The minimum number of cells in such products should be at least 7 log_10_ CFU/mL or g for starter bacteria or at least 6 log_10_ CFU/mL or g for labeled additional microorganisms (including probiotic strains) [15].

### 3.2. Changes in pH during the Fermentation Process

On the basis of the results and the final pH value of the fermented bean-based beverages obtained from germinated beans (Figure 1), the tested strains can be differentiated into two groups. The first group includes most of the strains used in this study (*Lactobacillus helveticus* LH-B01, *Lactobacillus paracasei* BGP1, *Lactobacillus plantarum* DSM 9843, *Lactobacillus rhamnosus* LH32, *Lactobacillus acidophilus* La3, *Lactobacillus brevis* L342, *Lactobacillus casei* 01, and *Lactobacillus delbrueckii* subsp. *lactis* ATCC 4797). This group is characterized by a significant decrease in the pH value from the initial value of 6.5–6.3 to the final value of 4.7–3.7. The second group includes two strains: *Lactobacillus fermentum* ATCC 9338 and *Lactobacillus delbrueckii* subsp. *bulgaricus* ATCC 11842. For this group, a much smaller reduction of the pH value was observed compared to that for the first group, with the final value reaching 5.8–5.6 (Figure 1). Statistical analysis showed that the strain used for fermentation was significantly (*p* < 0.05) associated with the final pH value achieved.

Because the strains we examined were derived from different species of lactobacilli, we observed differences in the acidifying abilities of different species of the *Lactobacillus* genus. On the other hand, literature data indicate a high dependence of the acidifying ability of bacteria on the strain of *Lactobacillus*. Gan et al. [16] used the *Lactobacillus plantarum* WCFS1 strain to ferment the beverages obtained from soybeans and mung beans. After 24 h of fermentation at 37 °C, the pH value decreased from 6.6 to approximately 3.5 in both types of beverages. On the other hand, Limon et al. [17] fermented a kidney bean beverage using the *Lactobacillus plantarum* ATCC 14917. After 48 h of fermentation at 37 °C, the pH value decreased from 6.60 to 3.76.

It should be emphasized that the acidifying abilities of the lactobacilli discussed in this work refer to the use of carbohydrates present in the bean seeds. The use of oligosaccharides present in beans by some lactobacilli is due to the enzymatic activity of these microorganisms. Bean seeds, like the seeds of other legumes, contain oligosaccharides from the raffinose family, which are α-galactosides. Many strains of *Lactobacillus* have the ability to ferment α-galactosides [9,18,19].

In our study, one of the weakest acidifying strains in this study were *Lactobacillus delbrueckii* subsp. *bulgaricus* ATCC 11842 and *Lactobacillus fermentum* ATCC 9338. In the literature, the *Lactobacillus bulgaricus* B548 strain was used to ferment a soybean beverage [20]. The pH value did not change after 24 h of fermentation and was 6.6. Only fermentation with the addition of 4% glucose decreased the pH value to 3.9. For comparison, *Lactobacillus fermentum* CRL 251 was used to ferment soybean beverages, and after 10 h of fermentation at 37 °C, the final pH value reached 4.5 [21]. The beverages made from peanuts and soybeans fermented by *Lactobacillus rhamnosus* LR32 after 24 h of fermentation at 37 °C reached the pH value of approximately 6 [22].

Our previous experiments [9] proved that the population of starter bacteria cells and probiotics in fermented beverages based on germinated beans was at the required level, i.e., at least 7 log_10_ CFU/mL and at least 6 log_10_ CFU/mL, respectively. The bacterial strains should therefore be properly selected in terms of specificity of the plant beverages due to the varying survival of lactobacilli [12]. The good viability of bacterial cells can be explained by the presence of oligosaccharides in bean-based beverages, which can be a carbon source for bacterial cells during fermentation [9].

### 3.3. Determination of the Fatty Acid Profile

The content of the lipid fraction in the raw bean seeds of white bean “Piękny Jaś Karłowy” and the bean-based beverages obtained from the germinated bean seeds of white bean was 1.18 ± 0.2 g/100 g and 0.16 ± 0.1 g /100 g, respectively. Table 2 shows the profile of selected fatty acids for the raw bean seeds of white bean “Piękny Jaś Karłowy” and the bean-based beverages obtained from the germinated bean seeds of white bean. In the fatty acid profile of raw beans, linolenic acid (C18:2n-6c) was the dominant fatty acid, whose share was 39.26% of the total fatty acids. Other unsaturated acids that were present in a significant share in the fatty acid profile included α-linolenic (23.25%) and oleic (17.59%) acids. The most abundant saturated acids were palmitic acid (12.79%) and stearic acid (3.68%). The remaining fatty acids had a share in the fatty acid profile in the level of approx. 0.5% and less.

Ryan et al. [5] reported the share of α-linolenic, linoleic, and oleic acids in the fatty acid profile of the kidney beans as 45.69%, 26.04%, and 11.97%, respectively. The proportion of palmitic acid and stearic acid was 14.2% and 1.3% of the total fatty acids, respectively. The proportion of unsaturated and saturated fatty acids was 83.8% and 16.5%, respectively. In another study, Grela and Gunter [23] determined the fatty acid profile for beans (*P. vulgaris*) and other legumes. For beans, the share of linoleic acid, α-linolenic acid, and oleic acid was 43.1%, 12.4%, and 13.9% in the fatty acid profile, respectively. Among the saturated fatty acids, the following acids had the highest share in the fatty acid profile: palmitic (16.8%) and stearic (3.5%).

The germination process modified the fatty acid profile of bean lipids. PUFA, namely linoleic acid (from 39.26 to 33.59% in the fatty acid profile) and α-linolenic acid (from 23.25 to 18.97% in the fatty acid profile), were found in the bean-based beverages obtained from the germinated bean seeds. The increase in the share of saturated fatty acids in the fatty acid profile, naming palmitic acid (from 12.79 to 16.72%) and stearic acid (from 3.68 to 5.66%), is worth noting. However, there was no change in oleic acid share in the fatty acid profile. 

The obtained results differ from that reported in the literature. Abdel-Rahman et al. [3] studied the influence of germination on the acid profile in mung bean lipids. The researchers, as in our study, observed an increase in palmitic acid share in the fatty acid profile (from 16.15 to 17.46%) and stearic acid share in the fatty acid profile (from 6.33 to 7.73%) as well as the decrease in linoleic acid share in the fatty acid profile (from 33.12 up to 30.56%). However, in contrast to our study, Abdel-Rahman et al. [3] showed a decrease in oleic acid share in the fatty acid profile (from 15.78 to 14.4%) and an increase in linoleic acid share in the fatty acid profile (from 16.98 to 18.34%).

It can be definitely stated that the fermentation process of bean-based beverages by various *Lactobacillus* strains decreases the proportion of palmitic acid and changes the pool of other detected fatty acids as compared to that observed in nonfermented beverages made from germinated bean seeds. Statistical analysis showed that the fermentation process significantly influenced the share of specific fatty acids in the fatty acid profile (*p* < 0.05).

The highest share of palmitic acid in the fatty acid profile was found in the bean-based beverages fermented by *Lactobacillus brevis* L342 (14.97% of this acid in the fatty acid pool), while the lowest share of palmitic acid in the fatty acid profile was found in the bean-based beverages fermented by *Lactobacillus plantarum* DSM 9843 (12.15% of this acid in the fatty acid pool). The highest share of stearic acid in the fatty acid profile was present in the bean-based beverages fermented by *Lactobacillus helveticus* LH–B01 (7.33% of this acid in the fatty acid pool). The bean-based beverages fermented by this strain was also distinguished by the highest total share of saturated acids in the fatty acid profile. The lowest share of stearic acid in the fatty acid profile was found in the bean-based beverages fermented by *Lactobacillus rhamnosus* LH32 (4.32% of this acid in the fatty acid pool). The highest share oleic acid in the fatty acid profile was found in the bean-based beverages fermented by *Lactobacillus casei* 01 (23.41% of this acid in the fatty acid pool). The lowest share of this acid in the fatty acid profile was found in the bean-based beverages fermented by *Lactobacillus fermentum* ATCC 9338 (17.52% of this acid in the fatty acid pool). The share of linoleic acid in the fatty acid profile was the highest in the bean-based beverages fermented by *Lactobacillus fermentum* ATCC 9338 (38.23% of this acid in the fatty acid pool), and the lowest share of this fatty acid in the fatty acid profile was observed in the bean-based beverages fermented by *Lactobacillus casei* 01 (32.99% of this acid in the fatty acid pool). The highest share of α-linolenic acid in the fatty acid profile was noted in the bean-based beverages fermented by *Lactobacillus delbrueckii* subsp. *bulgaricus* ATCC 11842 (22.42% of this acid in the fatty acid pool), and the lowest share of this fatty acid in the fatty acid profile was found in the bean-based beverages fermented by *Lactobacillus casei* 01 (19.05% of this acid in the fatty acid pool). In total, the highest share of unsaturated fatty acids in the fatty acid profile was found in the bean-based beverages fermented by *Lactobacillus delbrueckii* subsp. *bulgaricus* ATCC 11842 and the smallest share of these fatty acids in the fatty acid profile was observed in the bean-based beverages fermented by *Lactobacillus helveticus* LH-B01.

It should be noted that, in terms of the reported activity of lactobacilli, two strains that showed a weak fermentation capacity for the bean-based beverages (i.e., *Lactobacillus fermentum* ATCC 9338 and *Lactobacillus delbrueckii* subsp. *bulgaricus* ATCC 11842) did not differ significantly from the other *Lactobacillus* strains. This is in line with the work of Zaręba [24], who studied fatty acid profile of soya drink fermented by various bacteria strains of lactic acid fermentation and stated that yogurt starter culture had no statistically significant impact on the fatty acid profile during fermentation, and the only changes were observed during the refrigerated storage of the samples. 

It is known that lactic acid bacteria, including lactobacilli, possess an intracellular system of hydrolytic enzymes, especially lipases and esterases, which significantly contribute to the important transformation of lipids and fatty acids released in TAGs during the production of certain dairy products such as rennet ripened cheese [25,26,27,28]. Lactic acid bacteria have esterases and lipases that can hydrolyze a number of free fatty acid esters such as tri-, di-, and monoacylglycerols. It should, however, be noted that lipases and esterases of lactic acid bacteria are intracellular enzymes; therefore, the long maturation time and the subsequent lysis of cells allow their visible lipolytic activity during long-term maturation, which occurs in the production of ripened cheeses but is not observed in fermented beverages [29]. Bzducha and Obiedziński [30] showed that some strains of lactic acid bacteria from the *Lactobacillus* genus can also biosynthesize CLA by biohydrogenation. Akalin et al. [31] found that esterified forms of linoleic acid were also substrates of the CLA synthesis carried out by strain *Lactobacillus acidophilus* La–5 in yoghurts. Because of the metabolism of these bacteria, the content of the 18:2cis-9,trans-11 isomer in the tested products increased almost threefold. Therefore, it seems that such activity should also be observed in bean-based beverages. To our knowledge, there are no data on such activity on the fatty acids profile by lactobacilli, which may determine the change in the fatty acid profile during lactic acid fermentation of bean-based beverages. Our research indicates that this activity can be observed in at least some strains of *Lactobacillus*. Our research results are confirmed by the literature data. Pérez Pulido et al. [32] found several strains among lactobacilli with lipolytic activity, mainly heterofermentative lactobacilli strains (from *Lactobacillus brevis* and *Lactobacillus fermentum* species), although the detected activity was limited to short- and medium-chain fatty acid esters. Considering the low content of the lipid fraction in bean seeds and bean-based beverages obtained from them, the proportion of esterase activity is based rather on a limited hydrolysis of fatty acids. Moreover, Angeles and Marth [33] examined the lipolytic activity of dairy starter cultures in soy beverages and did not observe hydrolytic activity for the soybean lipid fraction by most of the tested strains. They, however, found little such activity in *Lactobacillus casei* and *Lactobacillus delbrueckii*.

### 3.4. Distribution of Fatty Acids in sn–2 and sn–1,3 Positions of Triacylglycerol (TAG) 

Not only the composition of fatty acids but also their positional distribution into particular positions of triacylglycerols influences specific nutritional properties of lipids. The enzyme responsible for eliminating individual acid residues, i.e., pancreatic lipase, functions only within external bonds (sn–1 and sn3), while acid residues located in the middle position (sn–2) are absorbed in the gastrointestinal tract in the form of monoacylglycerols, which is important from a nutritional point of view [34].

Figure 2 and Figure 3 show the positional distribution of fatty acids in the TAG of the raw bean seeds of white bean “Piękny Jaś Karłowy” and the obtained bean-based beverages. Statistical analysis of the obtained results showed that the positional distribution of fatty acids in the sn–2 position (*p* < 0.05) and in the sn–1 and sn–3 positions (*p* < 0.05) differed statistically significantly depending on the applied fermentation process.

In terms of the positional distribution of fatty acids in raw bean seeds, linoleic acid (C18:2n–6c) showed the highest share in the middle position (sn–2), which accounted for 46.53% of all fatty acids in this position. The other acids were α-linolenic (22.67%) and oleic (16.79%). Together, these three unsaturated acids had a share for almost 86% of all fatty acids contained in this position. Saturated acids, namely palmitic and stearic acids, accounted for 8.68% and 2.16%, respectively, of the total pool of fatty acids contained in the sn–2 position. In total, these two saturated acids had a share for 10.84% of all fatty acids contained in this position. Regarding the outer positions (sn–1 and sn–3), linoleic acid (39.53%) had the highest proportion, while stearic acid had the lowest proportion (19.55%). There was also a higher proportion of saturated acids and oleic and α-linolenic acids and a lower proportion of linoleic acid than those observed in the sn–2 position.

The obtained results differ from those obtained by other researchers, and this difference can be explained by generic, species, and even varietal variability of beans tested in the experiments. Yoshida et al. [11] showed such variability in the positional distribution of fatty acids in TAGs of kidney bean varieties. Linolenic acid dominated in the sn–2 position. It constituted 55–60% of the fatty acid pool in this position, depending on the variety of beans used. Linoleic acid accounted for approximately 20–30% of the fatty acid share in this position. Oleic, palmitic, and stearic acids were present in the amount of 1–10%. In the outer positions, the proportion of linolenic acid was 40–45% of the total fatty acid pool. The share of linolenic acid was 15–20%, and the palmitic acid share increased to approximately 20% of the total fatty acid pool in these positions [11]. Yoshida et al. [1] investigated the positional distribution of fatty acids in TAGs of Adzuki beans. The dominant fatty acid in the sn–2 position was linoleic acid, which constituted approximately 57–58% of the total fatty acid pool in this position, depending on the variety. α–Linolenic acid share in the fatty acid profile was at the level of 30–34%, while oleic (6.5–9.5%) and palmitic acids (2–3%) had a share in a much smaller level. Palmitic acid dominated in the outer positions (sn–1 and sn–3), which accounted for approximately 36% of the fatty acid share in these positions. This was followed by α-linolenic acid (23–28%), linoleic acid (approximately 19–20%), oleic acid (approximately 6–8%), and stearic (approximately 5–6%) [11].

Compared to raw beans, in the nonfermented bean-based beverages obtained from germinated beans, the share of palmitic acid increased from 8.68 to 16.54% and that of stearic acid increased from 2.16 to 5.64% in the sn–2 position (Figure 2). In general, the proportion of these two saturated fatty acids in the sn–2 position was greater than that in raw beans. The reduction of the share of polyunsaturated acids, namely linoleic acid and α-linoleic acid, is important, the share of which was 35.94% and 15.84%, respectively. The oleic acid share increased very slightly to the level of 17.13%. Regarding the share of the fatty acids in the outer positions (i.e., sn–1 and sn–3; Figure 3) in the nonfermented bean-based beverages obtained from germinated beans, the share of saturated fatty acids was very similar to that in the middle position: the share was 16.79% and 5.66% for palmitic acid and stearic acid, respectively. The highest share of linoleic acid (32.37%) was found in the outer positions. Linoleic acid had the highest percentage in the sn–2 position (35.70% of this acid was in this position). The smallest percentage was shown by α-linolenic acid (27.85% of the total share of this acid was in this position). Compared to raw beans, the changes that have occurred may have been due to biochemical and physical processes that occur during bean seed germination and the preparation of the bean-based beverages (i.e., cooking and sterilization). Unfortunately, the available scientific literature did not provide information on the influence of these processes on the positional distribution of fatty acids in triacylglycerols in legumes.

Currently, there is less research work on the subject of research on the influence of various biological and technological processes on the positional distribution of fatty acids in TAG. In one of the few such studies, the effect of microwave heating was assessed on the distribution of fatty acids in the hypocotyl TAGs of two soybean seeds. It was found that minor but statistically significant changes in the distribution of fatty acids occur after heating for 12 min or more. These changes are manifested by an increase in the percentage of palmitic acid in the sn–1 and sn–3 positions and a decrease in the percentage of linoleic acid in the sn–2 position [35].

The fermentation process also caused changes in the fatty acids share in the middle (sn–2) and outer positions (sn–1 and sn–3), and the differences in the proportion of individual fatty acids in these positions were statistically significant. Compared to the bean-based beverages made from germinated beans before its fermentation, the bean-based beverages after fermentation usually showed a lower share of palmitic acid (except for the beverage fermented with *Lactobacillus casei* 01) and of stearic acid in the sn–2 position. The fermentation process also increased the share of oleic acid in the sn–2 position as compared to that in raw beans and bean-based beverages obtained from germinated beans. Fermented bean-based beverages showed an average higher share of polyunsaturated acids, namely linoleic acid and α-linolenic acid, than that observed in the bean-based beverages obtained from germinated seed (Figure 2). Regarding the share of individual acids in outer positions (sn–1 and sn–3), the share of individual fatty acids changed depending on the strain used to ferment the bean-based beverages. The bean-based beverages fermented by *Lactobacillus delbrueckii* subsp. *bulgaricus* ATCC 11842 showed the highest share of unsaturated fatty acids in outer positions (sn–1 and sn–3; Figure 3). The lowest share of these acids in sn–1 and sn–3 positions was characteristic for the bean-based beverages fermented by *Lactobacillus helveticus* LH-B01. The highest share of polyunsaturated acids, i.e., linoleic and α–linolenic acids, was found in bean–based beverages fermented by *Lactobacillus fermentum* ATCC 9338 and *Lactobacillus paracasei* BGP1, respectively. It should also be noted that the two strains that showed the weakest fermentation capacity of the bean-based beverages (i.e., *Lactobacillus fermentum* ATCC 9338 and *Lactobacillus delbrueckii* subsp. *bulgaricus* ATCC 11842) did not differ significantly from the other *Lactobacillus* strains in terms of activity.

The obtained results show clear changes in the share of certain fatty acids in sn–2 and sn–1 and sn–3 positions and the proportion of individual acids in the sn–2 position, which is probably caused by the transesterification process carried out by the bacteria used in this study. Lipases can be specific for a particular fatty acid or more generally for a certain class of fatty acids. The positional or regiospecificity of lipases is defined as the ability of these enzymes to distinguish between the two outer positions (primary ester bonds) and the inner position (secondary ester bonds) of the TAG backbone. During the hydrolysis of triacylglycerols, sn–1,3-regiospecific lipases preferentially hydrolyze the sn–1 and sn–3 positions before the sn–2 position. Unfortunately, the results obtained in our experiments cannot be compared with literature sources, as there are no data on this subject in the available literature.

## 4. Conclusions

Subjecting bean seeds to biological processes such as germination and then obtaining fermented beverages from such a matrix can improve their nutritional value. In the available sources, there are few publications on the influence of germination and lactic acid fermentation processes on the fatty acid profile and positional distribution of fatty acids in bean lipids. 

Our research on bean-based beverages was obtained from the germinated bean seeds of white bean “Piękny Jaś Karłowy” (*P. vulgaris*); however, it proved that it is extremely important in this case to select the appropriate lactic acid bacteria strain used for fermentation. A high population of live lactobacilli cells (over 7 log_10_ CFU/g) and the significant decrease in the pH value to the value of 4.7–3.7 after the fermentation process proved that bean-based beverages are a good environment for activity of the lactobacilli. The bean fermentation process contributed to an increase in the share of saturated fatty acids (palmitic and stearic) and oleic acid in the fatty acid profile compared to that in raw bean seeds. This process simultaneously reduced the share of polyunsaturated acids (linoleic and α-linolenic) in the fatty acid profile. Even in the case of the two strains, *Lactobacillus fermentum* ATCC 9338 and *Lactobacillus delbrueckii* subsp. *bulgaricus* ATCC 11842, such changes were observed. 

The lactic acid fermentation processes significantly influenced the positional distribution of fatty acids in the middle position (sn–2) and in the outer positions (sn–1 and sn–3). The lactic acid fermentation process increased the share of oleic acid in the sn–2 position as compared to that in raw bean seeds and bean-based beverages obtained from germinated beans. Compared to the bean-based beverages obtained from germinated beans before its fermentation, the bean-based beverages after lactic acid fermentation also usually showed a lower share of palmitic acid (except for the beverage fermented with *Lactobacillus casei* 01) and of stearic acid in the sn–2 position. Regarding the share of individual acids in outer positions (sn–1 and sn–3), depending on the *Lactobacillus* strain used to ferment the bean-based beverages, two less fermenting strains, i.e., *Lactobacillus fermentum* ATCC 9338 and *Lactobacillus delbrueckii* subsp. *bulgaricus* ATCC 11842, did not differ significantly from the other *Lactobacillus* strains in terms of activity related to the positional distribution of fatty acids.

The obtained results of these experiments allow for directing further research on the use of the lactic fermentation process to modify the nutritional value and technological properties of legume seeds.

## Figures and Tables

**Figure 1 microorganisms-08-01348-f001:**
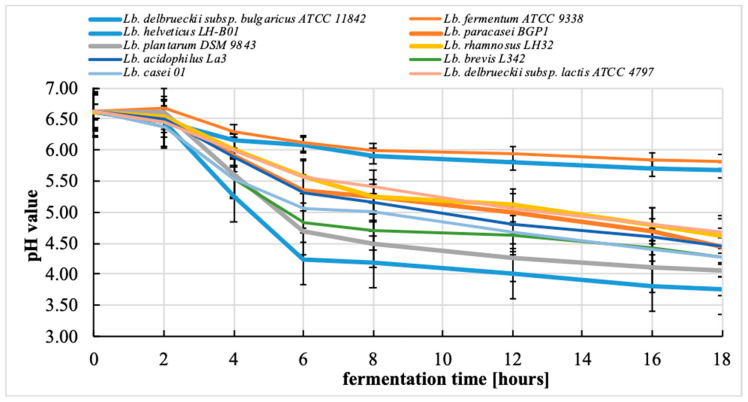
The pH value of bean–based beverages (means ± standard deviations, *n* = 4).

**Figure 2 microorganisms-08-01348-f002:**
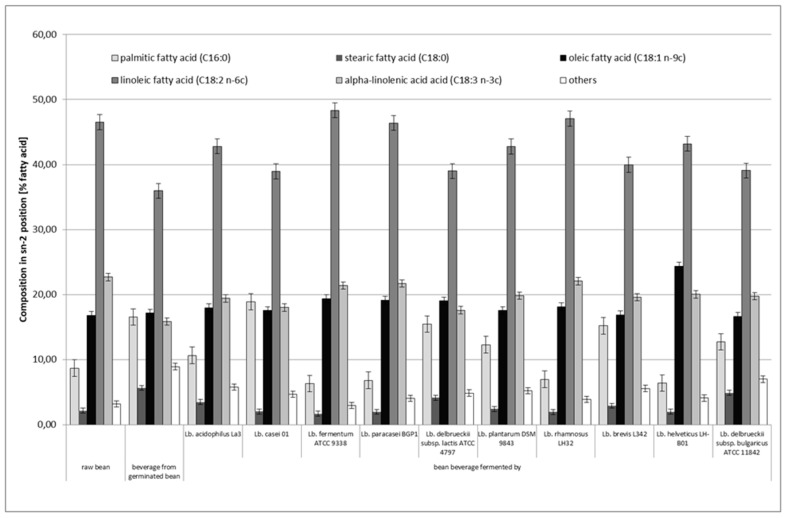
Positional distribution (sn–2) of fatty acids in triacylglycerols obtained from raw bean and bean–based beverages before and after fermentation process (means ± standard deviations, *n* = 4). Other minor fatty acids include 14:0, 16:1, 17:0, 20:0, and 22:0.

**Figure 3 microorganisms-08-01348-f003:**
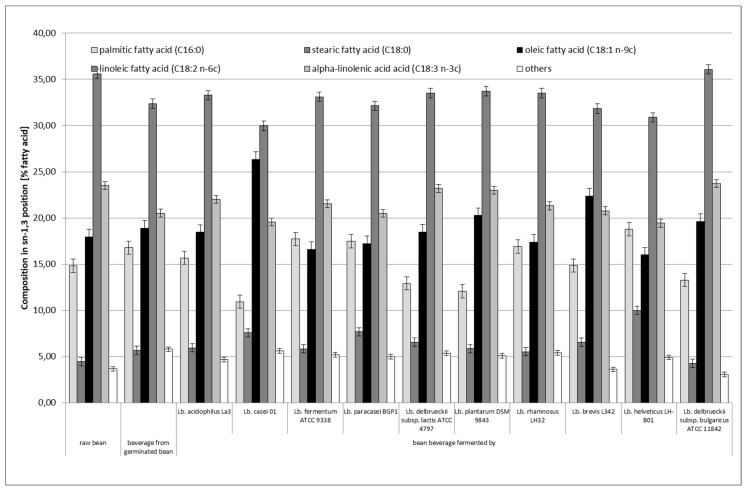
Positional distribution (sn–1,3) of fatty acids in triacylglycerols obtained from raw bean and bean–based beverages before and after fermentation process (means ± standard deviations, *n* = 4). Other minor fatty acids include 14:0, 16:1, 17:0, 20:0, and 22:0.

**Table 1 microorganisms-08-01348-t001:** Population of lactobacilli in bean–based beverages (means ± standard deviations, *n* = 4).

Bean–Based Beverages Fermented by:	Before Fermentation [log_10_ CFU/g]	After Fermentation [log_10_ CFU/g]
*Lb. acidophilus* La3	8.4 ± 0.3	8.3 ± 0.2
*Lb. brevis* L342	8.6 ± 0.2	8.2 ± 0.3
*Lb. casei* 01	7.6 ± 0.1	7.8 ± 0.4
*Lb. delbrueckii* subsp. *bulgaricus* ATCC 11842	7.5 ± 0.2	7.6 ± 0.2
*Lb. delbrueckii* subsp. *lactis* ATCC 4797	7.6 ± 0.3	7.5 ± 0.2
*Lb. fermentum* ATCC 9338	7.8 ± 0.4	8.1 ± 0.2
*Lb. helveticus* LH–B01	8.1 ± 0.2	8.0 ± 0.1
*Lb. paracasei* BGP1	8.2 ± 0.3	8.2 ± 0.3
*Lb. plantarum* DSM 9843	8.7 ± 0.2	8.5 ± 0.2
*Lb. rhamnosus* LH32	8.1 ± 0.1	8.2 ± 0.2

In all cases, there was no significant effect (*p* < 0.05).

**Table 2 microorganisms-08-01348-t002:** The fatty acids profile of the raw white bean “Piękny Jaś Karłowy” seeds and the obtained bean-based beverages (means ± standard deviations, *n* = 4).

Fatty Acid Share [%] in the Total Pool of Fatty Acids in the Samples:	Palmitic C16:0	Stearic C18:0	Oleic C18:1 n–9c	Linoleic C 18:2 n–6c	α–Linolenic C18:3 n–3
Raw white bean	12.79 ± 0.57 ^a^	3.68 ± 0.16 ^a^	17.59 ±0.78 ^a^	39.26 ±1.74 ^d^	23.25 ±1.03 ^c^
Bean–based beverages before fermentation	16.72 ± 0.74 ^d^	5.66 ± 0.25 ^c^	18.30 ±0.81 ^a^	33.59 ±1.49 ^a^	18.97 ±0.84 ^a^
Bean–based beverages fermented by:					
*Lactobacillus delbrueckii* subsp. *bulgaricus* ATCC 11842	13.10 ± 0.57 ^a.b^	4.46 ± 0.20 ^b^	18.62 ± 0.82 ^a^	37.12 ± 1.64 ^b.c^	22.42 ± 0.99 ^c^
*Lactobacillus fermentum* ATCC 9338	13.93 ± 0.62 ^b.c^	4.45 ± 0.20 ^b^	17.52 ± 0.78 ^a^	38.23 ± 1.69 ^c.d^	21.51 ± 0.95 ^b^
*Lactobacillus plantarum* DSM 9843	12.15 ± 0.54 ^a^	4.70 ± 0.21 ^b^	19.38 ± 0.86 ^b^	36.78 ± 1.63 ^b^	21.95 ± 0.97 ^b.c^
*Lactobacillus casei* 01	13.59 ± 0.60 ^b^	5.73 ± 0.25 ^c^	23.41 ± 1.04 ^d^	32.99 ± 1.46 ^a^	19.05 ± 0.84 ^a^
*Lactobacillus delbrueckii* subsp. *lactis* ATCC 4797	13.77 ± 0.61 ^b^	5.74 ± 0.25 ^c^	18.67 ± 0.83 ^a^	35.37 ± 1.56 ^a.b^	21.33 ± 0.94 ^b^
*Lactobacillus brevis* L342	14.97 ± 0.66 ^c^	5.34 ± 0.24 ^c^	20.54 ± 0.91 ^c^	34.58 ± 1.53 ^a^	20.40 ± 0.90 ^a.b^
*Lactobacillus helveticus* LH–B01	14.66 ± 0.65 ^c^	7.33 ± 0.32 ^d^	18.79 ± 0.83 ^a.b^	35.02 ± 1.55 ^a^	19.65 ± 0.87 ^a^
*Lactobacillus paracasei* BGP1	13.92 ± 0.62 ^b^	5.77 ± 0.26 ^c^	17.86 ± 0.79 ^a^	36.93 ± 1.63 ^b^	20.91 ± 0.93 ^a.b^
*Lactobacillus acidophilus* La3	14.00 ± 0.62 ^c^	5.11 ± 0.23 ^c^	18.28 ± 0.81 ^a^	36.49 ± 1.61 ^b^	21.16 ± 0.94 ^b^
*Lactobacillus rhamnosus* LH32	13.59 ± 0.60 ^b^	4.32 ± 0.19 ^b^	17.63 ± 0.78 ^a^	38.06 ±1.68 ^c^	21.67 ± 0.96 ^b^

Other minor fatty acids in the fatty acid profile include 14:0, 16:1, 17:0, 20:0 and 22:0. a, b, c—Means with different uppercase letters in the same column are significantly different (*p* < 0.05).

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
