# Peer review of "Effect of Lactic Acid Bacteria on the Lipid Profile of Bean-Based Plant Substitute of Fermented Milk"

_microorganisms, 2020, doi:10.3390/microorganisms8091348_

Round 1

Reviewer 1 Report

The work is interesting, however, the results are not of the appropriate quality. The main and serious drawback is the expression of fatty acids as “as share [%] in the total pool of fatty acids”. This does not allow to make any significant conclusions. In addition, I cannot find the total fatty acid content in mg/100g.

Abstract should be rewritten following the instruction for authors. Please see below: “Abstract: The abstract should be a total of about 200 words maximum. The abstract should be a single paragraph and should follow the style of structured abstracts, but without headings: 1) Background: Place the question addressed in a broad context and highlight the purpose of the study; 2) Methods: Describe briefly the main methods or treatments applied. Include any relevant preregistration numbers, and species and strains of any animals used. 3) Results: Summarize the article's main findings; and 4) Conclusion: Indicate the main conclusions or interpretations. The abstract should be an objective representation of the article: it must not contain results which are not presented and substantiated in the main text and should not exaggerate the main conclusions.” There is no conclusions in the present abstract.

Lines 41-42. I do not agree with that sentence. Several fermented foods are easily digestible and high in nutrients that are not vegan or only for vegetarians. In addition, the growing trend in these diets I do not think that is mainly to health eating or easy digestible.

Line 46. Please provide details on how the fermentation may increase the protein content of a food.

Line 57-87. This part does not fit with the aim of the present study, which is the effect of fermentation. In this part mainly the effect of germination is described. Please delete it and focus on studies with fermentation.

Introduction. Please clearly state the novelty of the present work and the question – problems solves.

Lines 114-115. I do not think that this temperature will stop all the actions by lactobacilli. There are numerous studies that show the activity of LAB even at these low temperatures. In addition, please provide details about cooling. For example how many time is required to reach 5 oC from 37? This will also affect the fermentation. Usually an ice bath is used to drop temperature faster.

Line 216. “did not change significantly (p > 0.005)”. Usually is p>0.05. Please explain. Also provide details in materials and methods for significance in p < 0.005. Please check it in the whole manuscript.

Table 1 Please correct , to . In addition, in my opinion the superscripts letters may be deleted and just state in the footnote of the table that there was no significant effect, in all cases.

Figure 1. The use of this line is not correct. It seems like a model was applied. Please add the values of each analysis and use a straight line between the analyses.

It was expected that L. bulgaricus would not reduce the pH. Why the authors did not use the traditional yogurt starter S. thermophilus and L. bulgaricus? This would lead to low pH. In addition, there are evidence that these starter may have a probiotic potential.

Lines 304-312. This part has no correlation with the present study. It should be deleted. In the present study % of total fatty acids is presented and no mg/100g.

Table 2. In my opinion, this table has no meaning and no conclusions can be made. We do not know of a specific fatty acids is reduced but only as share [%] in the total pool of fatty acids. Which is the total fatty acid pool in mg/100g. This would provide some explanation and quality in the results.

Lines 290-292. “The increase in the content of saturated fatty acids, naming palmitic acid (from 12.79% to 16.72%) and stearic acid (from 3.68% to 5.66%), is worth noting. However, there was no change in oleic acid content.” This is not an increase of the content. The palmitic acid may be the same and other acids just reduced. The results are % of total and not content. Please be careful and correct any part that may causes misunderstandings. The same applies in lines 313-316. The authors did not know the actual quantity of each acid.

Author Response

Author's Reply to the Review Report (Reviewer 1)

The work is interesting, however, the results are not of the appropriate quality. The main and serious drawback is the expression of fatty acids as “as share [%] in the total pool of fatty acids”. This does not allow to make any significant conclusions. In addition, I cannot find the total fatty acid content in mg/100g.

Response:

We thank the reviewer for appreciating our work. Thank you so much for all valuable comments

We studied the profile of fatty acids and therefore in the manuscript we used the expression of fatty acids as “as share [%] in the total pool of fatty acids”. To make this more clear, we have eliminated the phrases related to fatty acid content throughout the manuscript.

Comment:

Abstract should be rewritten following the instruction for authors. Please see below: “Abstract: The abstract should be a total of about 200 words maximum. The abstract should be a single paragraph and should follow the style of structured abstracts, but without headings: 1) Background: Place the question addressed in a broad context and highlight the purpose of the study; 2) Methods: Describe briefly the main methods or treatments applied. Include any relevant preregistration numbers, and species and strains of any animals used. 3) Results: Summarize the article's main findings; and 4) Conclusion: Indicate the main conclusions or interpretations. The abstract should be an objective representation of the article: it must not contain results which are not presented and substantiated in the main text and should not exaggerate the main conclusions.” There is no conclusions in the present abstract.

Response:

Abstract was rewritten following the instruction for authors and reviewers' suggestions.

Comment:

Lines 41-42. I do not agree with that sentence. Several fermented foods are easily digestible and high in nutrients that are not vegan or only for vegetarians. In addition, the growing trend in these diets I do not think that is mainly to health eating or easy digestible.

Response:

The suggestion has been incorporated in the revised manuscript. We changed the sentence:

“Current trends in the field of nutrition reflected in terms of search for easily digestible products that are rich in certain nutrients indicate the growing popularity of vegan and vegetarian diets. Modern technologies offer dairy-free products obtained from various plant raw materials. “

to:

“There are many fermented foods easily digestible and rich in nutrients. Several of them are suitable vegan and vegetarian diets. Modern technologies offer vegan or vegetarian products obtained from various plant raw materials.”

Comment:

Line 46. Please provide details on how the fermentation may increase the protein content of a food.

Response:

Of course, it is about increasing the digestibility of proteins, not their content.

Therefore the sentence:

“Subjecting bean seeds to biological processes such as fermentation may improve their nutritional value by increasing the content of certain nutrients (e.g., proteins or polyphenols) or by eliminating undesirable components (e.g., trypsin inhibitors, stachyose, and raffinose).

we changed to:”

“Subjecting bean seeds to biological processes such as fermentation may improve their nutritional value by increasing the content or digestibility of certain nutrients (e.g., proteins or polyphenols) or by eliminating undesirable components (e.g., trypsin inhibitors, stachyose, and raffinose).”

Comment:

Line 57-87. This part does not fit with the aim of the present study, which is the effect of fermentation. In this part mainly the effect of germination is described. Please delete it and focus on studies with fermentation.

Introduction. Please clearly state the novelty of the present work and the question – problems solves.

Response:

We have reduced this section of the manuscript to the one sentence only:

“An increasing number of studies is being conducted on the profile of fatty acids in seeds of various [1,3–8].”

Instead, we added some sentences state the novelty of the present work to the Introduction.

Comment:

Lines 114-115. I do not think that this temperature will stop all the actions by lactobacilli. There are numerous studies that show the activity of LAB even at these low temperatures. In addition, please provide details about cooling. For example how many time is required to reach 5 oC from 37? This will also affect the fermentation. Usually an ice bath is used to drop temperature faster.

Response:

We are aware that the temperature of 5oC does not stop the enzymatic activity of lactobacilli. Our other research also proves the activity of LAB even at low temperatures (there is a term called “cold fermentation” in the literature).

In this manuscript we meant their growth. In addition, we added information on the method of cooling samples after fermentation, therefore we corrected this sentence as follows:

“The inoculated samples were then incubated at 37°C for 18 h, and after completion of the fermentation process, the samples were cooled to 5°C by being kept in cold ice water for 30 minutes”.

Comment:

Line 216. “did not change significantly (p > 0.005)”. Usually is p>0.05. Please explain. Also provide details in materials and methods for significance in p < 0.005. Please check it in the whole manuscript.

Response:

It was an editorial error. We corrected it to p > 0.05 throughout the whole manuscript.

Comment:

Table 1 Please correct , to . In addition, in my opinion the superscripts letters may be deleted and just state in the footnote of the table that there was no significant effect, in all cases.

Response:

As suggested by the reviewer we deleted the superscripts letters and we added one short description below the table (In all cases there was no significant effect (p < 0.05)).

Comment:

Figure 1. The use of this line is not correct. It seems like a model was applied. Please add the values of each analysis and use a straight line between the analyses.

Response:

As suggested by the reviewer we modified the figure by connecting the points with straight lines. Moreover, we added the values of the standard deviation.

Comment:

It was expected that L. bulgaricus would not reduce the pH. Why the authors did not use the traditional yogurt starter S. thermophilus and L. bulgaricus? This would lead to low pH. In addition, there are evidence that these starter may have a probiotic potential.

Response:

We were sure that the tested Lactobacillus strains would reduce the pH value. This has already been shown to us by our earlier research. In other studies we used the traditional yogurt starters containing S. thermophilus and L. delbrueckii subsp. bulgaricus, and we obtained very positive fermentation results for the bean-based beverages. The traditional yogurt starters are great for fermenting a bean-based beverages, but starter cultures that also contain bifidobacteria and Lb. acidophilus are even better for this purpose (see https://www.vup.sk/en/index.php?mainID=2&navID=34&version=2&volume=58&article=2143).

In this study, we wanted to investigate the effect of pure cultures of single Lactobacillus strains. In the future, we also plan to investigate the impact of commercial starter cultures composed of many species, including yoghurt cultures and probiotic cultures.

Comment:

Lines 304-312. This part has no correlation with the present study. It should be deleted. In the present study % of total fatty acids is presented and no mg/100g.

Response:

As suggested by the reviewer we deleted this part of the manuscript.

Comment:

Table 2. In my opinion, this table has no meaning and no conclusions can be made. We do not know of a specific fatty acids is reduced but only as share [%] in the total pool of fatty acids. Which is the total fatty acid pool in mg/100g. This would provide some explanation and quality in the results.

Response:

We studied the fatty acids profile and therefore in the manuscript we used the expression of fatty acids profile as “as share [%] in the total pool of fatty acids”.

To make this more clear, we have eliminated the phrases related to fatty acid content throughout the whole manuscript.

Comment:

Lines 290-292. “The increase in the content of saturated fatty acids, naming palmitic acid (from 12.79% to 16.72%) and stearic acid (from 3.68% to 5.66%), is worth noting. However, there was no change in oleic acid content.” This is not an increase of the content. The palmitic acid may be the same and other acids just reduced. The results are % of total and not content. Please be careful and correct any part that may causes misunderstandings. The same applies in lines 313-316. The authors did not know the actual quantity of each acid.

Response:

We understand now that we have misused the term "content". We studied the fatty acids profile and therefore now we have improved the manuscript and we have eliminated the phrases related to fatty acid content throughout the whole manuscript.

Reviewer 2 Report

The production of food of animal production in energetically demanding and expensive. It seems that the ratio of animal products in human nutrition will be gradually decreased in the future. It is necessary to find adequate products of plant origin that ensure to replace meat, eggs, and dairy products in human nutrition. The ways to increase the suitability of plant-based foods is an actual and exciting research topic.

Authors Ziarno et al. submitted their manuscript entitled "Effect of lactic acid bacteria on the lipid profile of bean-2 based plant substitute of fermented milk "to Microorganisms.

I have several notices and recommendations:

L16-35: The abstract contains around 300 words. Is any reason for so extended abstract?

L22: CFU/g should not be in brackets.

L71: "... also studied also ...". Please, check the sentence.

L97-99: Did you use pure cultures of Lactobacillus strains from Chr. Hansen, or did you isolated them from dairy products?

L115: All these procedures ..." Please, check if the sentence is correct and logical. If conditions would be sterile (germ-free), you can't cultivate any bacteria. If you grow any bacteria, the conditions are not sterile. Please, rephrase the sentence.

L121-122: Which plates do you use for cultivation and which range of growing colonies per dish to you use for the counting.

L134: "the Folch method". Here should be a reference, e.g., https://link.springer.com/content/pdf/10.1007%2F978-94-007-7864-1_89-1.pdf

L153-161: If you did not develop this method, it should be supported by an appropriate reference.

L170-188: L153-161: If you did not develop this method, it should be supported by an appropriate reference.

L209: Did you test the normality of distribution before you chose the parametric test?

L209: Tukey's multiple comparisons post-hoc test?

L214: (CFU/g). The brackets should be removed. I will not pay to this repeated strange expression my attention more. Wouldn't it be better to express it as CFU/mL or CFU/L?

L216: P should by a capital letter and italic.

Author Response

Author's Reply to the Review Report (Reviewer 2)

Comments and Suggestions for Authors

The production of food of animal production in energetically demanding and expensive. It seems that the ratio of animal products in human nutrition will be gradually decreased in the future. It is necessary to find adequate products of plant origin that ensure to replace meat, eggs, and dairy products in human nutrition. The ways to increase the suitability of plant-based foods is an actual and exciting research topic.

Authors Ziarno et al. submitted their manuscript entitled "Effect of lactic acid bacteria on the lipid profile of bean-2 based plant substitute of fermented milk "to Microorganisms.

I have several notices and recommendations:

Response:

The authors thank the Reviewer for the detailed comments. We agree with most of the comments and the manuscript has been rewritten according to the comments.

We thank the Reviewer for appreciating our work.

Comment:

L16-35: The abstract contains around 300 words. Is any reason for so extended abstract?

Response:

Abstract was rewritten following the instruction for authors and reviewers' suggestions.

Comment:

L22: CFU/g should not be in brackets.

Response:

We have modified the notation to log10 CFU/g throughout the whole manuscript [see http://www.fao.org/3/a-i3996e.pdf].

Comment:

L71: "... also studied also ...". Please, check the sentence.

Response:

It was an editorial error. We corrected this sentence.

Comment:

L97-99: Did you use pure cultures of Lactobacillus strains from Chr. Hansen, or did you isolated them from dairy products?

Response:

In the case of strains from companies Chr. Hansen or Sacco we used pure cultures courtesy given from these companies, and we do not have to isolate them from dairy products. To make this information clearer we have modified it in the manuscript with the following sentence:

“The following strains of lactobacilli were used to obtain the bean-based plant substitute of fermented milk: Lactobacillus delbrueckii subsp. bulgaricus ATCC 11842, Lactobacillus delbrueckii subsp. lactis ATCC 4797, Lactobacillus fermentum ATCC 9338, Lactobacillus plantarum DSM 9843, Lactobacillus brevis L342 (from the WULS-SGGW pure culture collection), Lactobacillus acidophilus La3 (pure cultures from Sacco srl., Codarago, Italy), Lactobacillus paracasei BGP1 (pure cultures from Sacco srl., Codarago, Italy), Lactobacillus casei 01 (pure cultures from Chr. Hansen Poland, Czosnow, Poland), Lactobacillus rhamnosus LH32 (pure cultures from Chr. Hansen Poland), and Lactobacillus helveticus LH-B01 (pure cultures from Chr. Hansen Poland).”

Comment:

L115: All these procedures ..." Please, check if the sentence is correct and logical. If conditions would be sterile (germ-free), you can't cultivate any bacteria. If you grow any bacteria, the conditions are not sterile. Please, rephrase the sentence.

Response:

We meant aseptic conditions (free from contamination by microorganisms from the environment). We corrected this sentence.

Comment:

L121-122: Which plates do you use for cultivation and which range of growing colonies per dish to you use for the counting.

Response:

We used classic Petri dishes with a diameter of 9 cm. For colony counting, we chose Petri dishes in which the number of grown colonies was in the range of 30-300 colonies. This information was added to the manuscript:

“The Petri dishes with inoculations were incubated at 37°C for 72 h under anaerobic conditions using an anaerobic jar equipped with Anaerocult A (Merck KGaA). After the completion of incubation, the grown colonies were counted. For colony counting, the Petri dishes in which the number of grown colonies was in the range of 30-300 colonies were chosen. The final results were converted to the number of colony-forming units per gram of the product (CFU/g), and then converted to logarithm (log10 CFU/g).”

Comment:

L134: "the Folch method". Here should be a reference, e.g., https://link.springer.com/content/pdf/10.1007%2F978-94-007-7864-1_89-1.pdf

Response:

As suggested by the reviewer we added the indicated reference. And accordingly we have updated the references.

Comment:

L153-161: If you did not develop this method, it should be supported by an appropriate reference.

Response:

So far we have not used the MEGA-10 capillary column (before our experiments, were established and checked the column and method parameters, and of course we used the column according to manufacturer's recommendations), so we decided to describe all the parameters of the methods. MEGA-10 capillary column is equivalent to BPX-70, which is well known to us (see doi: 10.3390 / nu11091972).

Comment:

L170-188: L153-161: If you did not develop this method, it should be supported by an appropriate reference.

Response:

We used the methodology described in the papers: Yoshida, H.; Tomiyama, Y.; Mizushina, Y.; Characterization in the fatty acid distributions of triacyloglicerols and phospholipids in kidney beans (Phaseolus vulgaris L.). J. Food Lipids 2005, 12, 169–180 doi: 10.1111/j.1745-4522.2005.00016.x, and Yoshida, H.; Tomiyama, Y.; Yoshida, N.; Shibata, K.; Mizushina, Y.; Regiospecific profiles of fatty acids in triacylglycerols and phospholipids from adzuki beans (Vigna angularis) Nutr. 2010, 2, 49–59 doi: 10.3390/nu20100049, but with some modifications, so we decided to describe the entire methodology.

However, we have included a citation of the method used in the manuscript.

Comment:

L209: Did you test the normality of distribution before you chose the parametric test?

Response:

Of course, in each case we checked the normality of distribution before doing the parametric test. The obtained values of these statistics proved that our results have a normal distribution.

Comment:

L209: Tukey's multiple comparisons post-hoc test?

Response:

The purpose of Tukey's test is to check which groups in our samples are differ. Tukey's Test is common and popular method of post-hoc analysis (see https://en.wikipedia.org/wiki/Tukey%27s_range_test#Advantages_and_disadvantages).

Comment:

L214: (CFU/g). The brackets should be removed. I will not pay to this repeated strange expression my attention more. Wouldn't it be better to express it as CFU/mL or CFU/L?

Response:

We have modified the notation to log10 CFU/g throughout the whole manuscript [see http://www.fao.org/3/a-i3996e.pdf]. The convert CFU to log facilitates statistical analysis operations and not effect on microbiological results. Moreover, the logarithm transformation produces homogeneity of variance.

Comment:

L216: P should by a capital letter and italic.

Response:

As suggested by the reviewer we changed "p" to capital letter and italic "P" throughout the whole manuscript.

Round 2

Reviewer 1 Report

The authors have improved the manuscript following the reviewers' comments.

One comment

Line 56. I have concerns regarding the “increase of content”. Total phenolic content may increase or reduce due to interactions with protein but the content of protein, how it is possible to increase. Please delete increase or better clarify.

Author Response

Author's Reply to the Review Report (Reviewer 1)

Comments and Suggestions for Authors

The authors have improved the manuscript following the reviewers' comments.

One comment

Line 56. I have concerns regarding the “increase of content”. Total phenolic content may increase or reduce due to interactions with protein but the content of protein, how it is possible to increase. Please delete increase or better clarify.

Response:

We would like to thank the reviewer again for his thorough evaluation of our manuscript and for helping to improve it.

We changed sentence in the manuscript:

„Subjecting bean seeds to biological processes such as germination and fermentation may improve their nutritional value by increasing the content or digestibility of certain nutrients (e.g., proteins or polyphenols) or by eliminating undesirable components (e.g., trypsin inhibitors, stachyose, and raffinose).”

into:

“Subjecting bean seeds to biological processes such as germination and fermentation may improve their nutritional value by changing the content or digestibility of certain nutrients (e.g., proteins or polyphenols) or by eliminating undesirable components (e.g., trypsin inhibitors, stachyose, and raffinose).”